# Keyword-Conditioned Image Segmentation via the Cross-Attentive Alignment of Language and Vision Sensor Data

**DOI:** 10.3390/s25206353

**Published:** 2025-10-14

**Authors:** Hye Rim Kim, Byoung Chul Ko

**Affiliations:** Department of Computer Engineering, Keimyung University, Daegu 42601, Republic of Korea; 1114728@stu.kmu.ac.kr

**Keywords:** vision–language model, keyword condition, keyword-conditioned image segmentation, reasoning segmentation, multimodal learning, vision sensors

## Abstract

Advancements in multimodal large language models have opened up new possibilities for reasoning-based image segmentation by jointly processing visual and linguistic information. However, existing approaches often suffer from a semantic discrepancy between language interpretation and visual segmentation as a result of the lack of a structural connection between query understanding and segmentation execution. To address this issue, we propose a keyword-conditioned image segmentation model (KeySeg) as a novel architecture that explicitly encodes and integrates inferred query conditions into the segmentation process. KeySeg embeds the core concepts extracted from multimodal inputs into a dedicated [KEY] token, which is then fused with a [SEG] token through a cross-attention-based fusion module. This design enables the model to reflect query conditions explicitly and precisely in the segmentation criteria. Additionally, we introduce a keyword alignment loss that guides the [KEY] token to align closely with the semantic core of the input query, thereby enhancing the accuracy of condition interpretation. By separating the roles of condition reasoning and segmentation instruction, and making their interactions explicit, KeySeg achieves both expressive capacity and interpretative stability, even under complex language conditions.

## 1. Introduction

Recent advancements in large language models (LLMs) have enabled high-level reasoning tasks based on their powerful capabilities for natural language understanding and generation. Consequently, LLMs are being increasingly applied across a wide range of domains, including multimodal learning, natural language question answering, code generation, and visual recognition. In particular, vision–language models (VLMs), which jointly process linguistic and visual information, have emerged as a unified framework that bridges these two modalities. These models can perform complex tasks, including generating visual outputs such as object locations or segmentation masks, in response to textual queries.

Such VLM technologies have been extended to text-guided image segmentation, where a system identifies and segments specific objects in an image based on natural language descriptions provided by users. In this context, the image data are typically acquired using vision sensors such as RGB cameras, which serve as the fundamental source for object-level perception and segmentation tasks. This task requires the integration of various advanced techniques, including semantic alignment between language and vision, multimodal conditional reasoning, and object-level representation learning. Therefore, it is widely regarded as a representative benchmark for evaluating the true reasoning capabilities of VLMs.

Recently, several studies [1,2,3,4,5] have attempted query-based segmentation using multimodal large language models (MLLMs). These approaches typically enhance segmentation performance by incorporating special tokens such as the [SEG] token, visual attention mechanisms, and pixel-level language conditioning strategies. These VLM technologies have been actively applied to text-based image segmentation, where the system identifies specific objects within an image and generates corresponding segmentation masks based on natural language descriptions provided by users. This task inherently involves multiple challenging components, including semantic alignment between language and vision, multimodal conditional reasoning, and object-level representation learning, making it a representative benchmark for evaluating the true understanding capabilities of VLMs.

However, such models commonly suffer from interpretative discrepancies between linguistic responses and visual segmentation outputs. For example, even when a model generates a plausible textual response to a natural language query, it may fail to segment the corresponding object in the image accurately, either by omitting it entirely or by producing imprecise masks. This issue stems from a structural limitation, where the core concepts of the query are not explicitly reflected in the segmentation conditions. Consequently, there is often a lack of consistency between the language understanding and visual execution.

Figure 1 illustrates this problem. For the query ‘*Which object in this room is used for sleeping?*’, LISA [6] generates a reasonable textual response of ‘*The object in this room used for sleeping is a bed*’ but fails to produce a segmentation mask for the target object. In contrast, the proposed keyword-conditioned image segmentation model (KeySeg) generates both a coherent textual response and an accurate segmentation mask for the bed, demonstrating more effective task alignment between language and vision.

Existing segmentation models [7,8,9] typically incorporate natural language conditions either as part of the input context or through attention mechanisms that indirectly influence segmentation. However, these methods do not explicitly define the structural links between the linguistic conditions and the segmentation process. To address this limitation, we propose a framework that jointly interprets image and natural language queries, extracts keywords via multimodal reasoning, and utilizes them as explicit segmentation conditions, thereby enabling more precise and semantically aligned segmentation outputs. Existing vision–language segmentation models usually rely on a single special token that simultaneously conveys multiple roles. For example, a single token represents the conceptual meaning of the query or acts as the execution signal for segmentation. This implicit design often results in inconsistencies between the linguistic interpretation and the visual segmentation. In contrast, the proposed framework assigns these roles to two distinct tokens. The [KEY] token encodes the conceptual target inferred from multimodal reasoning over both the query and the image, while the [SEG] token is specialized solely for guiding the mask generation process. By disentangling these roles, KeySeg provides a clearer and more explicit alignment between condition interpretation and segmentation execution, addressing the limitations of previous approaches.

Specifically, we designed KeySeg to encode the inferred keyword information into a [KEY] token, which interacts with the [SEG] token responsible for generating the segmentation mask through a cross-attention-based fusion module. This module generates a semantically rich representation of the segmentation condition via [KEY] token embedding and precisely integrates it into the representation of the [SEG] token, thereby improving segmentation accuracy.

To ensure that the [KEY] token semantically captures the core concept inferred through multimodal reasoning, we introduce a keyword alignment loss that is integrated into the overall training objective. This loss encourages the [KEY] token to represent the intended conditional meaning derived from both the query and the visual context and enables the model to learn meaningful condition representations that directly influence segmentation decisions.

By explicitly leveraging conditional information and enforcing semantic alignment between the query and visual output, this architecture allows for more consistent and accurate segmentation, particularly in complex scenes or under linguistically challenging queries. The effectiveness of KeySeg was quantitatively validated through extensive experiments across multiple datasets. The main contributions of this work can be summarized as follows.

We propose a keyword-conditioned segmentation framework to resolve discrepancies in modality interpretation. The core concepts of natural language queries are inferred and embedded into a [KEY] token that is explicitly fused with the [SEG] token responsible for mask generation. This explicit integration bridges the gap between linguistic understanding and visual segmentation, thereby mitigating the discrepancies between the two modalities.We designed a cross-attention-based fusion module with keyword alignment loss. The fusion module facilitates semantic alignment between the [KEY] and [SEG] tokens, whereas keyword alignment loss encourages the [KEY] token to learn the intended semantics of the core concept inferred through multimodal reasoning from the query and the image. This approach improves the precision and effectiveness of condition-based learning.We construct a Reason-Seg-Key dataset and empirically validate the proposed model. Building on the existing ReasonSeg dataset, we augmented it with keyword annotations for each query to enable keyword-guided training. Experimental results on multiple datasets demonstrate that our model achieves superior accuracy and consistency compared with previous methods, validating the effectiveness of the proposed architecture.

## 2. Related Works

### 2.1. Multimodal Large Language Models

MLLMs typically perform text generation and reasoning based on visual context by integrating aligned image–language representations into LLMs. For example, BLIP-2 [10] employs a lightweight query transformer on top of a pre-trained vision encoder to project visual features into the LLM input space to delegate natural language generation to the LLM. Flamingo [11] introduces a perceiver re-sampler to summarize visual features, which are then incorporated into LLMs via cross-attention, enabling effective few-shot reasoning.

Models such as LLaVA [12] and MiniGPT-4 [13] align CLIP-derived visual features with LLM embeddings through linear projection and apply instruction tuning to specialize in tasks such as visual question answering and multimodal reasoning. Kosmos-1 [14] jointly trains images and text within a single transformer architecture to support end-to-end grounding, captioning, and OCR. Recently, GPT-4V [15] and Google Gemini [16] extended this framework to support generalized multimodal reasoning across diverse input and output modalities.

Although these MLLMs demonstrate impressive performance in language generation and high-level reasoning tasks, they remain limited to fine-grained pixel-level prediction tasks such as segmentation. A key limitation arises when natural language conditions are not consistently reflected in visual outputs, primarily because of the lack of explicit structural integration of conditional information into the segmentation process.

### 2.2. Referring Expression Segmentation

Referring Expression Segmentation is a task that segments a specific object in an image based on natural language expressions. It has been widely applied in various domains such as human–computer interaction, robotic vision, and augmented reality [17,18,19,20,21]. Early studies primarily focused on single-target settings, where each expression referred to exactly one object. Representative models such as MAttNet [22] and RRN [23] have implemented vision–language fusion architectures to segment the object referred to by explicitly referring to expressions. To address the complexity of real-world conditions, a generalized RES (GRES) model has been proposed as an extended setting in which a single expression may refer to multiple objects or none at all [24]. The gRefCOCO dataset was introduced to cover single-, multiple-, and no-target scenarios. GRES emphasizes the need for models that can jointly explain region-to-region and region-to-language alignments. For example, CGFormer [25] integrates a query-token-based mask classification framework with a cross-modal reasoning module to learn fine-grained correspondences between expressions and visual regions. Furthermore, it employs contrastive learning to reinforce object-level alignment and improve generalization to various forms of reference expressions.

Another major challenge in RES research is the limited availability of annotated training data. To address this issue, RESMatch [26], which is a semi-supervised learning framework, combines strong visual augmentation, text-based data augmentation, and pseudo-label refinement strategies, achieving performance comparable to that of fully supervised methods, even under weak supervision. Additionally, recent efforts have explored RES in zero-shot, weakly supervised settings. For example, global–local CLIP [27] leverages a pre-trained CLIP model to infer segmentation conditions by combining the global context with local phrase-level semantics, demonstrating strong generalization even with limited annotated data.

Despite these advancements, most existing RES approaches incorporate linguistic conditions indirectly through input embedding or attention mechanisms. Consequently, the segmentation criteria are not structurally encoded within the model, leading to frequent semantic misalignments between the language input and visual output.

### 2.3. Reasoning Segmentation

Reasoning Segmentation extends beyond simple object identification by requiring the interpretation of complex linguistic cues such as multiple conditions, relational descriptions, and conceptual classifications embedded in natural language queries. LISA [6] defines this task as ‘vision segmentation conditioned on high-level reasoning’, emphasizing that, unlike directive-based segmentation, it requires the model to understand and reason over the semantic structure and contextual relationships of the query prior to segmentation. In this context, models such as LISA and LLM-Seg [28] have attempted to leverage the language reasoning capabilities of LLMs to perform semantically grounded segmentation, even when queries lack explicit referring expressions. Reasoning-focused benchmarks such as ReasonSeg, MUSE [29] and LLM-Seg40K [28] have been introduced to evaluate segmentation under complex linguistic conditions and relational reasoning tasks.

However, a major limitation of existing approaches is their inability to handle segmentation conditions in a structurally explicit manner. Most models simply append query information to an input sequence or rely on attention-based mechanisms for implicit conditioning. Consequently, segmentation outputs often fail to align with the intended query semantics, leading to frequent interpretation discrepancies between language and visual responses.

To address this limitation, we propose a KeySeg architecture that structurally encodes key concepts inferred from multimodal reasoning. Specifically, we extract the core concepts from the query as a [KEY] token and explicitly align them with a [SEG] token, which is responsible for segmentation, through a cross-attention mechanism. By disentangling query interpretation and condition injection at the token level, our approach enables more accurate and consistent alignment between complex language inputs and visual segmentation tasks.

## 3. Method

### 3.1. Special Token Mechanism in MLLMs

MLLMs can utilize special tokens within their input sequences to achieve semantic distinction, assign roles, and inject specific conditions into inputs. In this study, as part of our special token design, we introduce the [SEG] and [KEY] tokens to convey the visual segmentation conditions to the model explicitly as linguistic expressions.

The [SEG] token carries an imperative meaning akin to ‘*generate a mask based on this point/region*’ and functions as a semantic query for the segmentation target object within the MLLMs. The model generates an embedding corresponding to this token as its output, which is subsequently used as a condition for segmentation mask generation by a dedicated mask decoder.

The [KEY] token encapsulates conceptual information about the segmentation target, as determined by the model’s comprehensive interpretation of both the image and natural language query. This mechanism differs from directly extracting keywords from the query. Instead, the model assesses the most suitable target based on the visual context and linguistic conditions, expressing this judgment as an embedding. The [KEY] token provides a dedicated representation of the core concept required to answer a query. Rather than relying solely on a literal keyword or a fully implicit embedding, it captures this concept through multimodal reasoning over both the query and the image. This design preserves the clarity of explicit conditioning while retaining the flexibility of implicit representations. Consequently, the segmentation process is guided by conditions that are both precise and semantically aligned. This [KEY] embedding is then used to refine and modulate the segmentation conditions through its semantic interaction with the [SEG] token.

The [SEG] and [KEY] tokens perform complementary functions and play a pivotal role in structuring segmentation conditions. Specifically, [SEG] operates as an execution criterion that informs the mask generation process, whereas [KEY] functions as a conceptual representation that defines the semantic criteria for segmentation. This explicit design of two functionally distinct tokens enables the model to construct a stepwise process for interpreting complex input conditions and inferring the corresponding segmentation target, thereby facilitating adaptable and context-aware segmentation performance across diverse scenarios.

This architecture promotes robust semantic alignment between visual and linguistic information, providing a strong foundation for the model to interpret intricate and composite conditions. Visual information is transformed into patch-wise embeddings using a CLIP-based image encoder [30], and textual inputs are tokenised and processed into language embeddings. The representations of each modality, along with the [SEG] and [KEY] tokens, are integrated into a single sequence and fed into the MLLMs. Based on this unified representation, the model identifies the target segmentation object according to natural language conditions and subsequently generates a precise mask for the corresponding regions.(1)$$F\left(x_{img},x_{txt}\right)=\left\{\left[SEG\right],\left[KEY\right],\ldots ,\left[EOS\right]\right\},$$

In this context, the [EOS] token indicates the end of a text sequence and delineates the overall input boundaries. This particular sequence composition is designed to go beyond mere information concatenation as it enables the model to learn the intricate relationships between various representations and execute segmentation judgments based on complex query conditions. Specifically, the reasoning segmentation task addressed in this study requires sophisticated condition interpretation and inferential capabilities. Therefore, a sequence design that structurally reflects the underlying meaning of a query is highly effective at enhancing both the model’s comprehension of the conditions and its segmentation accuracy. This special token-based input design provides structural flexibility within the MLLM for condition interpretation and segmentation execution and offers two key functional advantages.

Semantic Role Separation: The [SEG] and [KEY] tokens perform distinct and independent functions, representing the execution point for mask generation and the semantic criterion, respectively. This functional separation allows the model to differentiate and learn role-based representations clearly during training.Explicit Conditioning: The [KEY] token explicitly encodes conceptual information derived from the internal inference of the model. This mechanism ensures that the implicit linguistic meaning embedded within the segmentation conditions is effectively reflected and directly integrated into the input structure.

Unlike conventional approaches, this special token mechanism explicitly specifies and controls the conditional information at the token level, thereby simultaneously enhancing the interpretability of segmentation conditions and promoting the stability of representation learning.

### 3.2. Model Baseline

KeySeg is constructed based on pre-trained MLLMs and designed to interpret visual information and natural language queries holistically to perform precise object segmentation under complex conditions. We adopt LLaVA (Meta, Menlo Park, CA, USA) as the base multimodal large language model, which combines a CLIP vision encoder with an LLaMA-based language model. Our proposed [SEG] and [KEY] tokens and the fusion module are designed on top of this backbone. The overall model architecture is illustrated in Figure 2.

The input consists of an RGB image and a natural language query, which are transformed into visual patch representations using a CLIP-based vision encoder and language token embeddings using a tokenizer. These representations, along with the special tokens [SEG] and [KEY], form a unified input sequence that is fed into the MLLMs. This configuration encourages semantic alignment between the natural language condition and visual context, enabling the model to infer an object that corresponds to the query condition effectively.

The KeySeg MLLMs were constructed using pre-trained LLMs. To learn the semantics of the newly introduced special tokens effectively, we apply low-rank adaptation (LoRA) (Microsoft Research, Redmond, WA, USA) [31]. Because conventional pre-trained language models are not trained with special tokens such as [SEG] and [KEY], an additional process is necessary to teach them the functional roles of these tokens effectively, such as indicating segmentation criteria or expressing condition-centric representations. However, fine-tuning an entire large-scale model is computationally expensive and inefficient. Therefore, we adopted the LoRA method, which selectively trains only low-rank parameters while keeping most of the model’s weights frozen. This approach allows the model to recognize special tokens as new semantic units and acquire the representational capacity to utilize them appropriately in the multimodal condition interpretation process.

From the MLLM output sequence, the embedding corresponding to the [SEG] token is passed to a fusion module. This module refines the [SEG] representation to align precisely with the segmentation conditions through semantic interaction with the [KEY] token embedding. The corrected embedding is subsequently used as a core condition representation for the final segmentation judgment. This representation is then forwarded to a segment anything model (SAM)-based mask decoder [32] to generate a binary mask for the target object on a high-resolution visual feature map.

KeySeg is designed to separate query interpretation and condition injection structurally, and to define the semantic alignment process explicitly. This approach effectively mitigates the common issue of condition interpretation mismatch observed in existing LLM-based segmentation models, enabling the generation of high-precision segmentation results, even under complex natural language conditions.

### 3.3. Fusion Module

Among the MLLM output embeddings, the [SEG] token is semantically augmented through interaction with the keyword information provided by the [KEY] token before being transmitted to the mask decoder. To this end, a cross-attention-based fusion module was designed. This module enhances the interpretability of condition-based object segmentation by ensuring that the segmentation target concept inferred from the natural language query and represented by the [KEY] token is reflected more precisely and effectively in the [SEG] token.

The fusion module takes an MLLM output sequence as an input and performs a cross-attention (CrossAttn) operation. In this operation, the [SEG] token embedding serves as a query, whereas the [KEY] token embedding acts as both a key and a value. Through this process, the [SEG] token directly incorporates semantic conditional information from the [KEY] token, thereby refining its representation to become more condition-centric for the segmentation target. Each embedding is projected into a shared latent space to facilitate the attention operation, ensuring the seamless integration of keyword information into the [SEG] representation.

Following the cross-attention operation, the resulting output is added back to the original [SEG] embedding through a residual connection. This structural design enables the injection of precise conditional information while simultaneously preserving the original visual-context representation, ensuring both learning stability and the consistency of the learned representations. Furthermore, this approach allows the model to reflect new semantic information cumulatively without incurring information loss, thereby facilitating efficient training even in deep network architectures.

The refined [SEG] embedding subsequently undergoes a self-attention (SelfAttn) operation to learn the interactions with both visual and linguistic representations within the surrounding sequence. This self-attention mechanism dynamically learns contextual correlations, thereby enabling the [SEG] embedding to align effectively with the global visual context. Subsequently, this representation is further transformed into a final expression using a multilayer perceptron (MLP), which enhances the representational power of the model through nonlinear transformations in a high-dimensional space.

The entire process can be expressed mathematically as follows:(2)$$\hat{S}=MLP\left(SelfAttn\left(CrossAttn\left(S,K\right)+S\right)\right),$$
where S is the initial embedding of the [SEG] token, K is the [KEY] token embedding, and $\hat{S}$ is the final refined [SEG] embedding. The refined embedding $\hat{S}$ is then reordered to match the sequence length before being passed to the mask decoder (D), which serves as the input for generating the final object segmentation mask.(3)$$M=D\left(\hat{S}\right),$$

This structure enables the precise reflection of keyword-centric segmentation conditions, even in complex scenes or situations involving multiple objects, thereby contributing to the generation of more consistent and accurate segmentation masks by the model. With explicit condition injection facilitated by cross-attention, robust information preservation achieved through residual connections, and effective context alignment ensured by self-attention, all of which are designed in an integrated manner, this module plays a pivotal role in enhancing not only segmentation accuracy but also the reliability of condition interpretation.

### 3.4. Keyword-Conditioned Learning

In the proposed model, the [KEY] token is not merely used as a conditional input but is further extended to serve as an explicit learning signal. To this end, we define an auxiliary keyword prediction loss that is integrated into the overall training objective. This learning strategy encourages the model to reflect the semantics of the designated keyword for each training sample more precisely.

The keyword loss is computed as follows. First, we extract the embedding vector K corresponding to the [KEY] token from the output of the MLLMs. This embedding is passed through a linear projection and then compared with a vocabulary embedding table $E\in R^{V\times d}$ by computing the dot product to obtain the logits. Here V denotes the vocabulary size and d is the hidden dimension of the model. A temperature parameter τ is applied to modulate the output distribution and a cross-entropy loss is computed using the ground-truth keyword token y as the target. The complete formulation of keyword loss is defined as follows:(4)$$L_{keyword}=CrossEntropy\left(\frac{{W_{k}\cdot E}^{⊤}}{\tau },y\right),$$

Here, $W_{k}$ denotes the result of applying a linear transformation to the [KEY] token embedding K and E refers to the vocabulary embedding table. In addition to the keyword loss described above, KeySeg adopts a multi-objective loss framework to optimize various aspects of the model jointly, including language generation, segmentation accuracy, and condition expressiveness. Specifically, we include the following components:

Language generation loss $L_{CE}$, which preserves the language modeling capabilities of the MLLM;Binary cross-entropy loss $L_{BCE}$, which maximises the alignment between the predicted mask and ground-truth mask; andDice loss $L_{DICE}$, which further enhances segmentation consistency.

The total loss function for KeySeg is defined as follows:(5)$$L_{total}=L_{CE}+L_{BCE}+L_{DICE}+L_{keyword},$$

This multicomponent loss design balances different learning objectives while enabling the model to generate high-quality segmentation results that accurately reflect natural language conditions. The keyword loss term $L_{keyword}$ plays a crucial role in guiding the semantic representation of the [KEY] token. Rather than relying on a keyword as a simple input embedding, this loss encourages the model to infer and internalise the segmentation criterion through its reasoning process.

Consequently, KeySeg goes beyond basic condition-based segmentation by enabling the model to infer and represent segmentation criteria actively based on a broader linguistic context, ultimately supporting a more flexible and context-aware visual understanding.

## 4. Experiment

In this study, we evaluated the performance of our model for object segmentation conditioned on natural language queries using four benchmark datasets: refCOCO [33], refCOCO+ [33], refCOCOg [34], and ReasonSeg [6]. These datasets require models to generate segmentation masks based on the correspondence between visual content and natural language expressions. Each dataset varies in linguistic complexity and expression style, allowing a comprehensive assessment of a model’s reasoning capabilities across different semantic challenges.

### 4.1. Datasets

#### 4.1.1. The refCOCO Series

The refCOCO series is a large-scale vision language segmentation benchmark based on the MS COCO image dataset [35]. It focuses on expression segmentation tasks in which a natural language description is used to identify and segment a specific object in an image. The refCOCO dataset consists of 19,994 images, more than 50,000 annotated objects, and 142,209 reference expressions. These expressions tend to be short and direct, making the dataset well suited for evaluating basic referential grounding. For evaluation, refCOCO provides two separate testing splits to analyse performance differences based on target type and to control for visual and linguistic biases. Test A includes only expressions that refer to human subjects, whereas Test B includes expressions that refer to non-human objects. This split allows for a more detailed evaluation of a model’s ability to handle queries targeting people versus background- or object-focused queries, which may involve different levels of segmentation difficulty and visual attention requirements. refCOCO+ shares a similar structure with refCOCO but introduces an additional challenge by removing spatial terms from the referring expressions. This design prevents models from relying on explicit spatial cues, thereby requiring a deeper semantic understanding. The dataset contains 19,992 images, 49,856 objects, and 141,564 reference expressions. In contrast, refCOCOg consists of longer and more complex sentences, with a significant portion of queries involving ambiguity or relational descriptions. This dataset includes 25,799 images, 49,822 objects, and 95,010 reference expressions. Many queries require reasoning over the relationships between objects or an understanding of high-level contexts, making them particularly suitable for evaluating advanced language comprehension in segmentation tasks. Although these three datasets share the same image base as MS COCO, they differ significantly in terms of linguistic complexity and query construction. This allows for a multifaceted evaluation of a model’s ability to understand natural language and perform conditioned segmentation across various levels of semantic difficulty.

#### 4.1.2. ReasonSeg Dataset

ReasonSeg is a reasoning-based object segmentation dataset introduced in LISA and designed to go beyond explicit reference expression datasets. Unlike conventional datasets, where the target object is directly mentioned in the query, ReasonSeg presents queries that rely on indirect cues such as the object’s attributes, roles, usage, or contextual surroundings. The model must interpret both the image and query holistically to infer an appropriate segmentation target without an explicit reference. The dataset contains 1218 image–query–mask samples split into 239 training, 200 validation, and 779 testing examples. This setup provides an evaluation environment that assesses a model’s capability for high-level vision–language reasoning, rather than simple directive-based understanding.

While the refCOCO series is used to assess performance in explicit, directive-based segmentation, ReasonSeg focuses on evaluating a model’s reasoning-driven segmentation capabilities in more abstract and context-dependent language conditions. We conducted both quantitative and qualitative analyses across these datasets to evaluate the proposed model’s ability to perform accurate and consistent segmentation under varying levels of linguistic complexity and reasoning demands.

### 4.2. Model Training Using Reason-Seg-Key Dataset

The ReasonSeg dataset provides natural language queries and the corresponding object segmentation masks for images that often contain complex scenes. However, one limitation of this dataset is the absence of explicit annotations for key concepts or salient objects within each query, which requires the model to infer visual attention implicitly without guidance. To address this limitation, we constructed an extended version of the dataset called Reason-Seg-Key, which augments ReasonSeg with keyword-level supervision. The extended dataset retains the same number of samples as ReasonSeg but also includes automatically extracted keywords for each sample. These keywords are generated as single words or short noun phrases representing the core concept or referent of a query and its associated visual targets.

Keyword extraction is performed using the pre-trained FLAN-T5 [36] model with an instruction-style prompt such as ‘*Extract the most important object from the following text as a single word or short noun phrase*’. The resulting keywords are appended to the original dataset as a new field labeled ‘*keyword*’. To ensure the reliability of the automatically generated keywords, we combined simple post-processing with manual verification. Keywords that were incomplete or redundant were removed, and synonyms or multi-word expressions were standardized into a single representative form. Additionally, a subset of the extracted keywords was manually inspected and corrected as needed to ensure semantic consistency. By using this combination of normalization rules and manual checks, the dataset maintained consistent and reliable supervision signals across all samples.

In this study, we utilize the Reason-Seg-Key dataset for model training, treating the extracted keyword as the target for keyword-conditioned learning. Specifically, each keyword served as a supervised signal for training the [KEY] token, allowing the model to learn semantically grounded keyword-centric segmentation behaviors more effectively. By combining normalization rules and manual checks, the dataset maintained consistent and reliable supervision signals across all samples.

### 4.3. Experimental Setup

To evaluate the performance of the proposed keyword-conditioned segmentation model, experiments were conducted using three popular benchmarks for query-based object segmentation: refCOCO, refCOCO+, and refCOCOg. These datasets serve as standard benchmarks for expression segmentation, where the goal is to segment a specific object in an image based on a natural language query. Each dataset differs in segmentation difficulty, depending on the linguistic complexity and length of the referring expressions.

The refCOCO+ dataset presents a particularly challenging setup because of the presence of multiple similar objects in visually cluttered backgrounds, which makes it difficult to perform accurate segmentation based on text–image alignment alone. In contrast, refCOCOg contains relatively longer and more complex expressions, with an average of 8.4 words per query, increasing the overall reasoning difficulty. Therefore, refCOCOg is considered more challenging than refCOCO and refCOCO+, and strong performance on this dataset is indicative of the superior segmentation capabilities of a model under complex linguistic conditions.

Model performance was evaluated using two metrics. The centre-based intersection over union (cIoU) and generalised intersection over union (gIoU) assess the alignment between the predicted segmentation mask and ground-truth mask in terms of centrality and overall spatial consistency, respectively. For a fair comparison, all baseline models were evaluated using the same experimental pipeline. The results of LISA were reproduced using the official checkpoint and source code provided by its authors. The evaluations employed the same dataset splits and metrics used for KeySeg. The reproduced values may differ slightly from those reported in the LISA paper due to differences in preprocessing, metric implementation, and experimental environments.

### 4.4. Qualitative Evaluation

Table 1 presents the experimental results for the ReasonSeg dataset, where gIoU and cIoU were used as performance metrics to evaluate the proposed model’s inference-based object segmentation capabilities. ReasonSeg contains highly challenging queries that demand multistep reasoning and attribute inference without explicit instructions, making it suitable for assessing visual–linguistic integrated reasoning abilities beyond simple language understanding.

The experimental results reveal that KeySeg achieved an accuracy of 42.6% for the gIoU and 47.9% for the cIoU, outperforming all compared methods. Compared with LISA, our strongest competitor, KeySeg demonstrated significant performance improvements of 5.8% (gIoU) and 13.8% (cIoU), as well as exhibited a substantial performance gap when compared with other conventional segmentation models. These findings suggest that the proposed keyword-centric condition generation and semantic refinement approach offers a more robust representational structure for reasoning segmentation compared with simple mask condition injection methods or prompt-selection-based approaches.

The results empirically demonstrate that the proposed KeySeg architecture maintains consistent segmentation performance across various levels of linguistic complexity. Furthermore, our approach proves that keyword-based segmentation condition interpretation and representation refinement operate effectively, even on tasks requiring high-level reasoning. Notably, the incorporation of keyword loss learning enables the model to represent the central concepts inherent in the conditions accurately, providing strong alignment performance when converting MLLM outputs into segmentation criteria. Although this evaluation is based on a single dataset, ReasonSeg contains highly complex and reasoning-oriented queries, and the clear improvements of KeySeg under these conditions highlight the structural robustness of the proposed approach in handling reasoning-based segmentation tasks.

Table 2 presents a quantitative comparison of the proposed KeySeg model with existing methods on three standard benchmarks: refCOCO, refCOCO+, and refCOCOg. On the refCOCO dataset, KeySeg achieved cIoU values of 71.8% on Test A and 63.6% on Test B, outperforming all baseline models on both splits. This result demonstrates that the proposed keyword-conditioned architecture effectively facilitates precise object identification, even in scenarios involving relatively simple directive expressions. For refCOCO+, KeySeg achieved 63.1% accuracy on Test A and 46.4% accuracy on Test B. On the Test A split, KeySeg surpassed the previous best-performing VLT model (61.0%) by 2.1%. Although KeySeg’s performance on Test B is slightly lower than that of VLT, KeySeg still outperforms LISA and MCN by 4.6% and 3.7%, respectively. Although KeySeg performs slightly worse than VLT on the Test B split, this outcome reflects the characteristics of the data. Test B primarily consists of object-centric queries, where success relies more on robust object category recognition than on reasoning over semantic attributes or relations. VLT benefits from large-scale pretraining that enhances general object recognition and thus gains an advantage in this setting. In contrast, KeySeg demonstrates superior performance on Test A and refCOCOg, where queries demand higher-level reasoning beyond simple object identification. On the more challenging refCOCOg dataset, KeySeg achieved a cIoU of 62.4%, outperforming both LISA and VLT, and achieved the highest performance among all compared models.

Both refCOCO+ and refCOCOg pose significant challenges because of the removal of spatial cues and presence of longer and more complex queries, respectively. These settings require a higher level of linguistic understanding and reasoning beyond simple directive segmentation. The improved performance under these conditions highlights the effectiveness of the structural design of KeySeg, particularly in terms of capturing and utilizing semantic conditions for accurate object segmentation. In particular, the cross-attention-based fusion module between the [KEY] and [SEG] tokens enables the model to integrate the core concepts from a query into a semantically meaningful segmentation instruction. Additionally, the incorporation of keyword loss provides an explicit training signal that refines the model’s ability to represent and focus on the correct target object with greater precision.

### 4.5. Ablation Study

#### 4.5.1. Effect of KEY Token Insertion

Table 3 presents the results of an ablation study conducted on the refCOCOg dataset to evaluate the impact of the [KEY] tokens on segmentation performance quantitatively. For this experiment, we adopted a LISA-based architecture as the baseline, where only the [SEG] token was inserted into the multimodal input. Since LISA itself is implemented on top of LLaVA, keeping the same backbone ensures a fair comparison. Therefore, all improvements shown in Table 3 can be attributed to our proposed [KEY] token and the fusion mechanism, rather than differences in the underlying language model.

As an intermediate variant, we first experimented with a configuration that included only the [SEG] token but enhanced its contextual understanding by integrating it with a self-attention-based fusion module. This setting achieved 59.89% gIoU and 58.78% cIoU, representing improvements of 2.2% and 1.71%, respectively, over the baseline. These results suggest that enriching the [SEG] representation through self-attention-based fusion partially contributes to improved context-aware condition modelling.

The best performance was achieved when both [SEG] and [KEY] tokens were incorporated, and a fusion module combining cross-attention and self-attention was used to enable bidirectional interaction. This configuration achieved 63.85% gIoU and 62.40% cIoU, representing improvements of 6.16% and 5.33%, respectively, over the baseline. In this setup, key semantic concepts were explicitly extracted from the query and encoded into the [KEY] token. This information was then fused into the [SEG] token through the designed attention mechanisms, enabling the model to interpret the intended segmentation criteria more precisely and align them with the generated mask.

This structural enhancement is particularly effective under the high linguistic complexity of refCOCOg, where queries are often long and semantically rich. The explicit decomposition of query semantics and integration of these cues through cross-attention allow the model to produce more consistent and accurate segmentation results by aligning core meanings with visual outputs.

These ablation results empirically demonstrate that keyword-guided conditioning and cross-attentive fusion modules are key architectural components for structurally improving referring expression segmentation. Although extending the ablation to multiple datasets would provide stronger validation, we focused on refCOCOg because it best represents the reasoning-oriented scenarios targeted in this study. We recognize the importance of broader multi-dataset analysis and regard it as a key direction for future work.

#### 4.5.2. Fusion Module Configuration Analysis

Table 4 presents a quantitative analysis of the effects of the internal components of the fusion module on the conditional alignment and segmentation performances on the refCOCOg dataset. This ablation study compared the three configurations based on the inclusion of cross-attention, self-attention, and residual connections. The first configuration applies a simple cross-attention mechanism in which condition-centric information from the [KEY] token is directly injected into the [SEG] token without any internal interaction or feedback. This unidirectional information flow limits the ability of the model to interpret complex or lengthy queries because the [SEG] token cannot holistically incorporate the full structure of the input query. Consequently, this setting achieved 56.51% gIoU and 55.83% cIoU, suggesting that under such conditions, the segmentation output suffers from insufficient semantic alignment.

The second configuration enhanced the previous setup by introducing self-attention into the [SEG] token, allowing the [SEG] representation to interact with the entire input sequence, helping it better contextualise the condition signal and suppress interference from irrelevant background objects or non-target elements. This configuration aims to improve the selectivity and consistency of condition interpretation, particularly in complex scenes with multiple objects or where spatial and attribute-level reasoning is required. However, it exhibited a slightly lower performance (55.41% gIoU and 55.57% cIoU) than the cross-attention-only setting. This result suggests that although self-attention can enrich contextual understanding, it may also dilute the model’s focus on the core semantic signal conveyed by the [KEY] token. The final configuration combines cross-attention and self-attention, and introduces a residual connection that aggregates the fused condition representation back into the original [SEG] embedding. This design aims to preserve the semantic precision of the condition representation while gradually reinforcing the alignment throughout training. This configuration achieved the best performance, with 63.85% gIoU and 62.40% cIoU, demonstrating robust and consistent segmentation results, even under ambiguous or linguistically complex queries.

These results indicate that simply adding individual attention mechanisms is insufficient. The interaction between attention types and the overall flow of information within the fusion module plays a crucial role in enhancing multimodal condition understanding. Notably, the final fusion design proved to be both semantically reliable and structurally effective for keyword-based alignment and segmentation instructions, providing empirical validation of its robustness under complex natural language conditions.

### 4.6. Quantitative Evaluation

Figure 3 visually compares the segmentation results of LISA and our KeySeg model on the refCOCOg dataset. refCOCOg is characterised by the frequent presence of multiple similar objects, complex backgrounds, and lengthy queries, demanding both precise language–visual alignment and accurate object distinction. In this comparison, we analysed the performance of the proposed model by focusing on three major difficulty scenarios: multi-object coexistence, distinction between similar objects, and background information interference.

First, in instances where multiple similar objects are present within the frame (e.g., vehicles in the second row, mountain goats in the third row, and cyclists in the fourth row), standard segmentation models often suffer from selection errors or the erroneous inclusion of multiple objects, despite the query specifically indicating a single target object, as a result of the high visual similarity among objects. Indeed, LISA was observed to include multiple objects or generate blurred masks that failed to distinguish the designated target accurately. In contrast, by leveraging its complementary special token structure, KeySeg successfully segmented only a specified single object by precisely reflecting the meaning of the query. This outcome demonstrates that the keyword fusion module effectively incorporates semantic differences between objects into its representation.

Furthermore, situations in which complex background elements overlap with the main object or similar shapes in the background interfere with object identification (e.g., the kitchen scene in the first row and the swan and water reflection in the last row), highlighting the critical role of a model’s representational power in terms of segmentation boundary consistency and accuracy. KeySeg effectively suppresses visual interference factors and extracts clear contours of the designated object. Conversely, LISA exhibits a lack of consistency, including either the background around the object or incorrect segmentation of the reflected shapes. This result indicates that a keyword-based refinement structure contributes to condition-centric representation reconstruction beyond simple visual features.

These qualitative comparison results demonstrate that the proposed model is capable of performing semantic-based representation alignment, even under complex and ambiguous visual–linguistic conditions. It can accurately infer and segment only conditioned objects, even in scenarios where multiple or similar objects are present. In particular, an integrated design combining a cross-attention-based fusion structure with keyword-aware learning is crucial for effectively isolating semantic differences between objects and simultaneously ensuring the stability and precision of segmentation performance.

## 5. Conclusions

In this paper, we proposed KeySeg, a keyword-conditioned image segmentation model designed to mitigate the performance discrepancies often observed in MLLM-based object segmentation. The proposed KeySeg model was engineered to incorporate semantic-centric information into the segmentation process by extracting keyword information from natural language queries using [KEY] tokens and fusing them with [SEG] token embeddings through a dedicated fusion module. Furthermore, we introduced keyword loss as an explicit learning signal for the [KEY] token, enabling the model to learn conditional information clearly. Experimental results demonstrated that the proposed model outperforms existing baseline models, thereby validating the effectiveness of our keyword fusion module and conditional learning structure in terms of enhancing object segmentation accuracy.

Future work will focus on strengthening both the robustness and interpretability of the proposed framework. Beyond refCOCOg, evaluation on multiple datasets with diverse linguistic complexities would provide stronger evidence of generalizability. In parallel, a more fine-grained analysis of the loss design in Equation (5) could reveal how different objectives contribute to stabilizing [KEY] token learning and overall segmentation quality. Finally, richer qualitative comparisons will be explored to illustrate how KeySeg handles ambiguous queries, failure cases, and subtle differences across models, thereby offering deeper insights into its reasoning process.

## Figures and Tables

**Figure 1 sensors-25-06353-f001:**
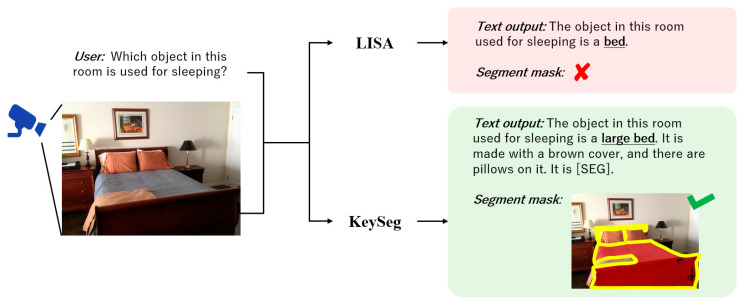
An example of cross-modality task interpretation discrepancies using image data acquired from vision sensors. While LISA generates an appropriate language response to a natural language query, it fails to segment the corresponding object visually. In contrast, the proposed KeySeg model produces both a consistent textual response and an accurate segmentation mask for the same query.

**Figure 2 sensors-25-06353-f002:**
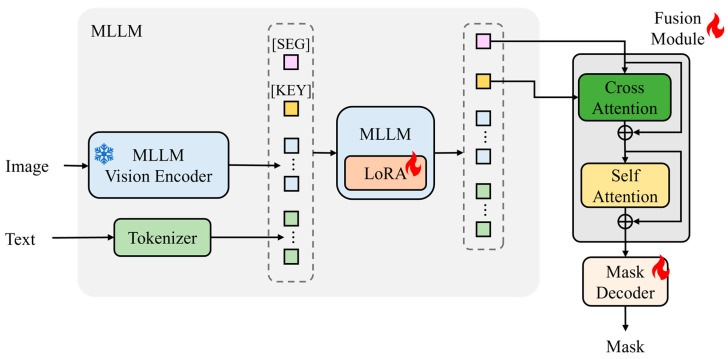
Overall architecture of the proposed KeySeg model. The input image and query are processed through a vision encoder and tokenizer, and then passed to the MLLMs with [SEG] and [KEY] tokens indicating the segmentation condition. The [SEG] and [KEY] tokens are newly introduced and optimized during fine-tuning. At inference, the mask decoder uses the refined [SEG] embedding enriched via cross-attention with the [KEY] token. Unlike LISA, which implicitly combined query interpretation and segmentation in a single token, KeySeg separates these roles into two tokens explicitly connected through the fusion module.

**Figure 3 sensors-25-06353-f003:**
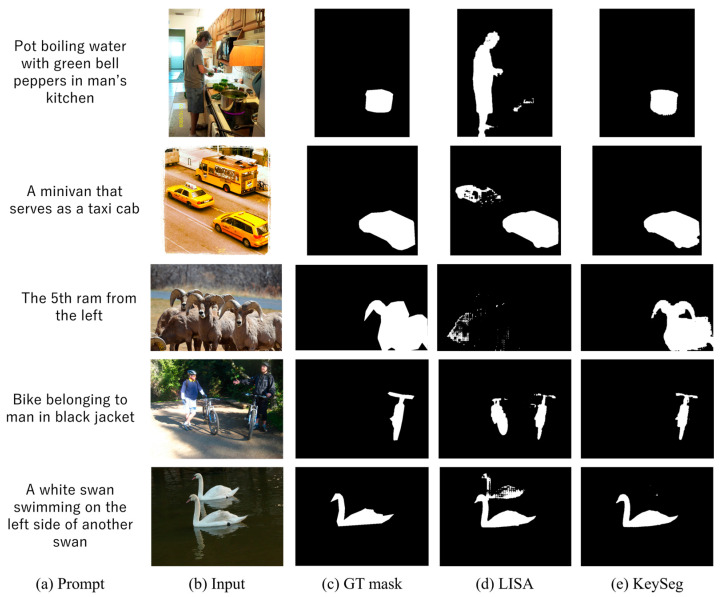
Comparison of object segmentation results on the refCOCOg testset.

**Table 1 sensors-25-06353-t001:** Evaluation on the ReasonSeg Dataset.

Method	ReasonSeg	
	gIoU	cIoU
OVSeg [37]	26.1	20.8
GRES [25]	21.3	22.0
X-Decoder [38]	21.7	16.3
SEEM [39]	24.3	18.7
Grounded-SAM [40]	21.3	16.4
LISA [6]	36.8	34.1
**KeySeg**	**42.6**	**47.9**

**Table 2 sensors-25-06353-t002:** Evaluation on rerfCOCO series datasets (cIoU).

Method	refCOCO		refCOCO+		refCOCOg
	Test A	Test B	Test A	Test B	Test
MCN [41]	64.2	59.7	55.0	42.7	49.4
VLT [42]	70.5	65.2	61.0	**50.1**	57.7
LISA [6]	68.7	59.8	56.9	41.8	57.0
**KeySeg**	**71.8**	**63.6**	**63.1**	46.4	**62.4**

**Table 3 sensors-25-06353-t003:** Ablation study on [SEG], [KEY], and the Fusion Module on RefCOCOg. CA and SA denote cross-attention and self-attention, respectively.

[SEG]	[KEY]	Fusion Module		refCOCOg	
		CA	SA	gIoU	cIoU
✓				57.69	57.07
✓			✓	59.89	58.78
✓	✓	✓	✓	**63.85**	**62.4** **0**

**Table 4 sensors-25-06353-t004:** Ablation study on attention mechanisms and residual connection design in the fusion module.

CA	SA	Residual Connection	refCOCOg	
			gIoU	cIoU
✓			56.51	55.83
✓	✓		55.41	55.57
✓	✓	✓	**63.8** **5**	**62.4** **0**

## Data Availability

Data are contained within the article.

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
