# Peer review of "Keyword-Conditioned Image Segmentation via the Cross-Attentive Alignment of Language and Vision Sensor Data"

_sensors, 2025, doi:10.3390/s25206353_

Round 1
Reviewer 1 Report
Comments and Suggestions for Authors
The authors present their novel method of KeySeg for keyword-conditioned image segmentation and show performance improvement over SOTA methods. But, adding special tokens for downstream applications is a well-known approach, and so the impact of the method would be limited to the specific application of keyword-conditioned image segmentation.
The methods presentation needs improvement.
- What is the baseline of the proposed method?
- 'keyword' does not always capture the full meaning of 'query'. May need comparison between explicit keyword extraction and implicit semantic representation.
- Figure 2 is not clear. How are [SEG] and [KEY] initialized during inference? Does Mask decoder use only [SEG] and [KEY] embeddings? How is it different from SOTA methods?
The evaluation results require further elaboration.
- Table 1: This is the key evaluation results, but is based on a single dataset. The performance improvement over the SOTA method LISA is impressive, but it's unsure if it would be robust.
- Table 2: Why does VLT outperform KeySeg for testB in both datasets of refCOCO and refCOCO+?
- Table 3: Only one dataset. Need to be done for multiple datasets.
- Need ablation study about the multiple components of loss function in Eq. (5)
Author Response
Please refer the 'Reviewer Reponse1' file. Thanks.

Reviewer 2 Report
Comments and Suggestions for Authors
This paper proposes KeySeg, which is a keyword-conditioned image segmentation framework to overcome the limitations of multimodal large language models. The method introduced two special tokens and a cross-attention-based fusion module, combined with a keyword alignment loss. Experimental results on three datasets show improved segmentation accuracy.
The paper is well written and clearly structured, and addresses an interesting problem in multimodal learning. The novelty of the method lies in explicitly encoding inferred query conditions through the [KEY] token and aligning it with segmentation criteria. The detailed results on three datasets and ablation studies are convincing. However, there are a few points that need clarification.
1) Introduction highlights the limitations of existing approaches; it is recommended to explicitly write the distinction of KeySeg relative to LISA and other recent vision-language segmentation models.
2) The keyword extraction using FLAN-T5 is briefly mentioned. How was keyword quality controlled (manual verification or automatic)?, Were synonyms or multi-word phrases handled consistently?
3) Which base MLLM is used for the KeySeg model? The base MLLMs should be mentioned, and their effect on performance.
4) There is a discrepancy in the results. The reported results of LISA are higher in their paper. Please mention why they are different. If authors fine-tuned or evaluated those methods by themselves, please write details about them.
5) Qualitative comparison with other methods from Tables 1 and 2, and on other datasets, would enhance the paper.
Author Response
Please refer the 'Reviewer Reponse2' file. Thanks.

Round 2
Reviewer 2 Report
Comments and Suggestions for Authors
Dear Authors, Thank you for providing clarifications. All my concerns have been resolved, and I would suggest accepting the paper in its current form.